# Digital Authoritarianism: Protecting Islam in Multireligious Malaysia

Syaza Shukri

Department of Political Science, International Islamic University Malaysia, Kuala Lumpur 50728, Malaysia; syazashukri@iium.edu.my

**Abstract:** Mahathir Mohamad's autocratic leadership over Malaysia for decades has left the country with a reputation for being, at best, a hybrid system. The country witnessed the rise of the internet during Mahathir's first term as prime minister, which led to the establishment of the Malaysian Communications and Multimedia Commission (MCMC) in 1998 to oversee telecommunications and the internet within the country. Since then, the MCMC has overseen the removal of inflammatory content from the internet. The Malaysian government has emphasised its commitment to purging the internet of harmful content including pornography, gambling, and offensive teachings about Islam in the name of safeguarding the religion and its adherents. Since the 1980s, Islam has been institutionalised in Malaysia, and the government has also used the faith as rationale for policing online behaviour especially on the 3R—religion, race, and royalty. With the cover of religious rhetoric like preventing "fitnah," or social upheaval, the government has used Islam to legitimise its activities in curtailing free expression online, including criticism of the government. Recently, Islam has also been utilised by populist actors in their online posting with little repercussions. This article explains the methods by which the Malaysian government has attempted to limit web access using religious discourse as justification. Since Malaysia has a Muslim majority, such restrictions can be justified in the name of Islam at the expense of the minorities.

**Keywords:** digital authoritarianism; Islamist populism; Malaysia; Malay; media censorship

## 1. Introduction

Malaysia is an emerging democracy with elections being held since before independence in 1957. However, for 61 years, Malaysia has had a dominant party system in which the UMNO-led Barisan Nasional was the government of the day. During this period, opposition was suppressed under various mechanisms such as the colonial-era Internal Security Act (ISA). Malaysia was thus considered an illiberal democracy at worst, and a hybrid regime at best. In 2018, the Pakatan Harapan coalition defeated Barisan Nasional in the general elections and for the first time Malaysia experienced a peaceful transition in government. Unfortunately, the Pakatan Harapan experiment lasted for 22 months without completing its full term before the administration reverted to the previous elites and establishment in a soft coup in which members of parliament from Bersatu, a Pakatan Harapan component party, decided to leave the coalition. With increasing contestation, Islamist populism has risen within the online space and thus further threatens Malaysia's move towards democracy due to its exclusivist nature. Since the formation of the so-called "backdoor government" in 2020, Malaysia has dropped in ranking in various indexes from Freedom House, Corruption Perception Index, and World Press Freedom Index. Although these indexes are inherently biased and western-centric, they do provide a qualitative sense of where Malaysia was heading despite lacking accuracy.

After winning the 2018 general elections, Pakatan Harapan attempted some reforms including the repeal of the Anti-Fake News Act in December 2019, a law which would exert huge fines for alleged fake news or partly fake news. However, other problematic laws such

as the Sedition Act stay in place. Opponents fear the Sedition Act would be used against political opponents but given the rise of dangerous populist rhetoric, Malaysia may need to keep the Act for the foreseeable future. Based on a national survey by Malaysia's own think tank, Merdeka Center, they found that a sizable majority of respondents believe that media reporting in Malaysia is biased. While non-Malay respondents are more receptive to press freedom, a sizable majority of respondents support anti-fake news measures despite worries that the authorities would abuse them (Merdeka Center 2021).

The change in government in 2020 also coincided with the COVID-19 pandemic which caused severe restrictions on the media. Freedom House (2021a) noted the decline in score on media freedom was "due to increased government pressure on private media, including law enforcement actions in response to critical coverage and the government's use of the COVID-19 pandemic as a pretext to prevent independent outlets from covering key events" (Section D, para. 4). Increasingly, the rise of populism also explains current digital authoritarianism in Malaysia where "identarian populism, another form of religious populism, has arisen. It uses religion to demarcate a civilizational division between "the people" and "the Other."" (Yilmaz et al. 2022, p. 10) In Malaysia, it is the Muslim majority against the rest.

Table 1 shows that Malaysia has been on an upward trend when it comes to internet freedom, in line with Malaysia's rising score in global freedom. While the country saw a massive increase in the global freedom score from 45 in 2018 to 52 in 2019 following the electoral defeat of Barisan Nasional, the trend on freedom on the net has been relatively stable with gradual increases over the years. In fact, it is fair to say that Malaysians enjoy greater freedom online than they do offline as it is difficult to fully censor the internet now that the people have experienced relative ease of access. It is also worth noting that despite the 2020 soft coup, the succeeding Perikatan Nasional government has maintained a relatively open space for dissension online. This has its pros and cons as in other countries. While an open space allows for open discourse and debate, it also allows for fringes in society to spread misinformation and disinformation.

**Table 1.** Malaysia's score on internet freedom 2016–2021.

| | Obstacles to Access | Limits on Content | Violation of User Rights | Total Score |
|---|---|---|---|---|
| 2016 | 16 | 19 | 20 | 55 |
| 2017 | 17 | 19 | 20 | 56 |
| 2018 | 17 | 19 | 19 | 55 |
| 2019 | 17 | 21 | 19 | 57 |
| 2020 | 17 | 21 | 20 | 58 |
| 2021 | 17 | 21 | 20 | 58 |

Source: Freedom on the Net 2021: Malaysia (Freedom House 2021b).

This article is interested in unpacking the means by which the Malaysian government has tried to limit access online by using religious rhetoric as justification. The argument is that since Malaysia has a majority Muslim population, it is easier to justify such restrictions in the name of Islam even if it jeopardizes the harmony of the diverse population. Interestingly, this has resulted in Islamist populists coming into cyberspace by utilizing the same religious language adopted by the government. The following section will firstly discuss the literature on democracy and internet restriction in order to provide a context for Malaysia. Next, we will discuss the various methods of government restrictions online and finally the role of religion—specifically Islam—in this digital authoritarianism.

## 2. Democracy and Internet Censorship in Malaysia

The "third wave" of global democratisation, which started in the 1970s and culminated following the end of the Cold War, saw the rise in cyber activity and access to the internet. Unfortunately, a considerable "democratic retrenchment" globally has marked the first two

decades of the twenty-first century ([Diamond 2015](#)). This current wave also coincided with the rise of populist leaders ([Kyle and Meyer 2020](#)) which further damaged long-established democracies that were earlier assumed to be immune.

The populist wave has evolved over time, diversifying as well as growing. At least three broad types of populism exist today ([Kyle and Gultchin 2018](#)). Anti-establishment is the most well-known type, where members of the "political elite" and other groups are vilified as part of a populist narrative. Socioeconomic populism is a center-left philosophy that seeks wealth redistribution to the greater population. The final and most pervasive group is cultural populism, which portrays individuals and groups both inside and outside the nation as "the adversary" of the "pure people." Malaysia is currently experiencing the rise of cultural populism in the form of Islamist populism ([Shukri and Smajljaj 2020](#)).

When Islam is politicised by populist leaders, it not only serves as a means of separating "the pious people" from "the corrupt secular elite," but also employs a religious symbolism and style in which adherents are urged to live out explicit religious and conservative lifestyles. On its own it is not a problem. The issue arises when those who do not succumb to the idea are delegitimized within the political space and worse, may become victims of abuse online and offline. These populists have been at the forefront of political messaging in the internet sphere.

As a result, the early digital optimism of the 1990s has given way to a type of digital pessimism, or at the very least cynicism, in a world where democracy appears to be "in retreat." There is rising fear that semi-authoritarian governments may utilise digital tools to establish their control even though individuals can use social media and digital technology to keep informed and participate in online and offline action more democratically. Thus, the call for action towards the government is in response to the rise of this anti-democratic populist forces that may lead to polarization in Malaysia.

Southeast Asia as a region is known to have strict authoritarian governments since the 1960s supposedly in the name of preserving national peace. Over the decades, alongside economic developments, countries such as Malaysia have adopted more democratic practices albeit with rampant money politics and relatively weak institutions. The government's reach is thus not limited to the 'real world' but with the spread and growth of internet connectivity, Malaysia's authoritarian government has tried to limit political participation online as well.

The Malaysian government is not much different from its neighbours in dealing with online political activities. Malaysia's political opposition was forced online following the 1998 *reformasi* movement because they were denied access to mainstream media. As a result, they turned to "alternative" media online such as using blogposts. The Malaysian government attempted to target political activists in cyberspace by using laws already in place for maintaining racial harmony and national security, such as the Sedition Act and the Communications and Multimedia Act. However, the government saw an abrupt response from the online community, who accused the former of violating the people's fundamental rights to freedom of speech. As a result, the government changed its approach to combat reformer political activism, especially after the government lost its traditional two-thirds majority in the 2008 election, by using the same means to spread its own narratives ([Thein 2012](#)). Additionally, it assigned cybertroopers to attack the opposition reformists and their discourses in accordance with that strategy.

[Johns and Cheong](#) ([2019](#)) looked at the Malaysian government's response to the *Bersih* street protests that emerged in 2007. According to the analysis, the authorities used two different strategies to stifle the protest movement. First, it sent out cybertroopers to spread articles that would harm the protest movement's reputation. These cybertroopers circulated statements saying things such as that *Bersih*'s primary goal was to undermine the nation's political stability, and was therefore also against the religion of the people—Islam. Second, the government implemented the Sedition Act and the Communications and Multimedia Act. It detained several bloggers and social media activists on the grounds that their posts endangered the political and social stability of the nation. These regulations prompted a

phenomenon known as "self-censorship", where people were compelled to stop engaging in political discourse on social media platforms.

In the early days of internet activism, Ficher (2009) found that despite the Malay middle class's use of the internet for political activism and to voice their concerns about government policies, attempts were made by the administration to muzzle criticism by monitoring political websites. The government's strategy, however, was at odds with its desire to participate in the global economy through fostering unrestricted digital communication. Interestingly, despite government coercion, Malaysians have a chance to come together online as a nation to confront present socio-political difficulties by seeing beyond racial and ethnic barriers (Wok and Mohamed 2017). The general public constantly looks for new, secure options, such as private WhatsApp and Facebook groups, to get back into political discussions (Johns and Cheong 2019). The government uses coercive tactics that encourage self-censorship among Malaysians but with the widespread use of social media platforms such as TikTok, it has become almost impossible to limit access to information online and as a result populist rhetoric has been used by the government and private citizens.

### 3. Internet Governance Institutions

Conventional media in Malaysia have typically been linked to the various ruling parties as their mouthpieces. They are also tightly controlled by the Printing Presses and Publications Act 1984 which governs the different facets of print publication. This act, however, could not be used to police electronic media and this led to the rise of blogs and online news portals in the second half of the 1990s. As part of Mahathir Mohamad's agenda to develop the internet and other information and communication technologies (ICT), a 10-point promise was made including a guarantee of no censorship of the internet (MSC Status Office 2017). This is also guaranteed through Section 3(3) of the Communications and Multimedia Act 1998 (CMA), which states that "Nothing in this Act shall be construed as permitting the censorship of the Internet." However, in the decades since, there has been increased censorship, blocking of websites, surveillance, and regulation of political and social expression.

The main body that regulates telecommunications and the internet in Malaysia is the Malaysian Communications and Multimedia Commission (MCMC) established in 1998. The Communications and Multimedia Act 1998 provided the regulatory licensing framework to the media and telecommunication industry whereas the MCMC Act created the MCMC itself. There is a Communications and Multimedia Content Forum to govern content on the internet based on a voluntary content code, but the Forum is relatively ineffectual. Over the years, MCMC has been accused of doing the government's bidding in shutting down supposedly aberrant websites and pursuing those who are critical of the government in cyberspace.

Section 211 and Section 233 of the CMA 1998 makes it illegal for platforms and users to share or create content that is "indecent, obscene, false, menacing, or offensive in character with intent to annoy, abuse, threaten or harass any person". The problem lies with the definition of the words "offensive" and "annoy". Under Section 213 of the CMA 1998, offensive content is "anything which offends good taste or decency; is offensive to public feeling, is likely to encourage crime or lead to disorder, or is abusive or threatening in nature" based on "the country's social, religious, political, and educational attitudes and observances, as well as the need to accommodate global diversity in a borderless world." While the MCMC does not technically block websites, Section 263(2) of the CMA 1998 requires Malaysian Internet Service Providers (ISPs) to assist MCMC in preventing the above criminal offences. Section 242 of the CMA 1998 provides the punishment for failure to comply with MCMC which is a fine not exceeding RM100,000, or prison not exceeding two years, or both.

In 2017, the Malaysian National Cyber Security Agency (NACSA) was established under the National Security Council "with the objectives of securing and strengthening Malaysia's resilience in facing the threats of cyber attacks, by co-ordinating and consol-

idating the nation's best experts and resources in the field of cyber security." (NACSA n.d.) Although the aim is to consolidate efforts surrounding cybersecurity, the widened definition of cybersecurity in Malaysia has threatened the role of civil society as a voice of dissent against the government. Internet regulation in the name of cybersecurity has been used to restrict internet freedom amongst political and social dissidents such as in the creation of the Anti-Fake News Act in 2018, which is the first of its kind among Southeast Asian countries. (Leong and Lee 2021) The framework for this study will be based on a report published by the European Centre for Populism Studies (Yilmaz et al. 2022) that described four levels of internet governance The following subsections discuss these four levels of governance separately to distinctly identify various types of censorship and digital authoritarianism.

### 3.1. Full Network Level Governance

Unlike countries such as Turkey (e.g., 2016 internet closure following the arrests of Diyarbakir's mayor), India (e.g., in Jammu and Kashmir in 2019) and Pakistan (e.g., internet blackout in 2012 to discourage Ashura procession by Shiites), the Malaysian government has yet to fully shut down the internet. The closest the government came to being accused of affecting the full network was in 2012 when mobile phone usage was disrupted during the *Bersih* civil rally (Yuen 2012). Even then, the government claimed that the slow internet speed was simply due to more users being online than on foul play by the government. In general, Malaysia suffers from a relatively low mobile internet speed at 31.34 Mbps in September 2021 (Speedtest 2021). For comparison, the global average is 63.15 Mbps. Malaysia does fare better for fixed broadband, ranking at 46th place with a speed of 107.55 Mbps. The slow internet connection was exacerbated during the lockdown period in 2020 and 2021 due to COVID-19 as more people went online while being forced to stay at home. Another reason given in 2016 was that the government was more focused on providing greater coverage across the country than on increasing the internet speed (Malay Mail 2016). Malaysians would enjoy 5G technology in Malaysia in 2023 after the government postponed its initial plan for implementation (Adilla 2020). Overall, the Malaysian government has been committed to maintaining and improving full network capabilities in the country.

### 3.2. Sub-Network or Website Level Governance

Although the Malaysian government has never shut down the internet, it has actively blocked thousands of webpages that fit the bill of the "offensive" definition according to the CMA Act 1998. Malaysia has even received a score of 7 out 11 by Comparitech, a UK tech website, with a score of 11 denoting fully censored internet in a country (Moody 2021). Based on data from OONI Explorer, multiple websites are censored in Malaysia ranging from pornography to websites that criticised Islam. MCMC would block websites when they received complaints and applications from government ministries and agencies such as the Royal Police (PDRM), Health Ministry, Tourism Ministry and Domestic Trade, Cooperatives and Consumerism Ministry. From 2018 until 2020, 2921 pornographic websites were blocked while 4277 pornographic websites were blocked from 2015 until 2016 (Malik 2021; *Bernama* 2016b). In relation to this, following a widely reported infographic on sugar babies among Malaysian students in 2021, MCMC blocked the Sugarbook website for violating Section 233 of the CMA Act 1998 due to having elements of prostitution (Rozaidee 2021). Beyond its criminal element, the lifestyle of sugar babies among university students were widely condemned by Malaysians who are generally more conservative and look down upon such activities.

Similarly, pornography sites are among the most blocked websites by MCMC as they go against the norm and sensitivity of the majority Muslim population in Malaysia. Based on Section 292 of the Penal Code, Malaysians are not allowed to possess anything pornographic in nature. A high-profile case in 2013 involved two persons who were charged for posting pornographic images of themselves on their blog (*The Edge Markets*



2014). However, there is no law against simply watching porn online, but as discussed previously, MCMC is active in blocking such websites in order to abide by the expected customs of Malaysian society.

Furthermore, 2195 online gambling websites have been taken down until March 2021 (MCMC 2021b). MCMC restricts access to these websites in order "to keep Internet users safe" (*Bernama* 2016a). What is actually meant is that the contents in these websites go against the social norms of the conservative population of the country; also, some websites have possible criminal elements to them. At the height of ISIS rule in the Middle East, MCMC also blocked 72 websites in relation to the spread of extremist ideology in 2015 (*Bernama* 2016b).

MCMC has also been accused of abusing its power against the opposition. During the 14th general election in 2018, MCMC ordered 11 internet service providers to block three websites by *Malaysiakini*, a popular online news portal known for its neutrality (and thus not being a proxy for the government), on live updates of the election results for fear it could affect "national stability, public order and harmony, and economic stability" (*Malaysiakini* 2018).

*3.3. Proxy or Corporation Level Governance*

In comparison to other countries known for their strict internet access such as Turkey and Pakistan, the Malaysian government has been less restrictive towards corporations, including social media companies. At the same time, MCMC had requested social media platforms such as Facebook, Twitter and YouTube to take down contents that go against the social and cultural norms of Malaysians such as gambling, in accordance with the terms of service and the community standards. According to MCMC, 97.3% of internet users own a Facebook account, making it the most popular social networking site, whereas 98.1% of users prefer WhatsApp as a communication app (MCMC 2018).

According to Twitter's Transparency Report (2020), there were 275 legal demands to remove or withhold content on the platform from 2012 until 2020 by the Malaysian authority. Interestingly, out of the total, 153 requests, or 55.6%, were made in the period of July until December 2020, a huge spike from previous years right after the change of government as discussed above. An example of such request was in December 2020 against *Bermana TV*, a parody account that satirized the *Bernama* news agency.

The same trend is observed according to Facebook's Transparency Center (2020). 376 contents were restricted by Facebook between January and June 2020, more than double the previous count at 163 from July until December 2019. The pandemic has provided an opportunity for the Perikatan Nasional government to clamp down on critics under the guise of combating fake news. This is done through the Emergency (Essential Powers) (No.2) Ordinance 2021, which was enforced throughout the Emergency Ordinance period from January until July 2021. The majority of the content restriction in the second half of 2020 was made on Facebook posts (10), followed by pages and groups (7) and profiles (5). According to the report, during this period in 2020, Facebook restricted access in Malaysia to 10 items reported by the Malaysian Communications and Multimedia Commission (MCMC), including 5 items pertaining to COVID-19 misinformation that violated Penal Code Sections 505(b) and 124I, and 3 items that were alleged to constitute locally illegal hate speech. Facebook also restricted access to two items which violated CMA Section 233(1)(a).

Another method that the government has used to manage proxies is by prosecuting online news portals. For example, the Federal Court found in February 2021 that the online news portal *Malaysiakini* is liable for contempt of court over five readers' comments that were alleged to have "clearly meant that the judiciary committed wrongdoings, is involved in corruption, does not uphold justice and compromised its integrity." (*Malaysiakini* 2020) This was following the amendment made in 2012 to the Evidence Act 1950 which allowed for repercussions on online commentors. The main controversy over the insertion of Section 114A into the Act is that anyone who "facilitates" the publication of offending material,

even though not being the person behind the comments, such as social media organizations, online forums, news webpages, or even public places that provide WiFi, may be liable to legal action. Although *Malaysiakini* was fined an exorbitant RM500,000—beyond the RM200,000 sought by the prosecution—they were able to crowdsource the fund within a few hours after the judgement.

### 3.4. Network-Node or Individual Level Governance

Of all the levels of network governance, the prosecution and harassment of users are the more common method for the Malaysian government to control the internet. As discussed in the literature review, the government decided to use the internet to its advantage instead of trying to shut it down. This has impacted Malaysia's ranking according to the French-based Reporters Without Borders as Malaysia dropped 18 spots in 2021 after improving its positions for two consecutive years previously (RSF 2021). In 2020, under the Perikatan Nasional government, six journalists from Al Jazeera were investigated for alleged sedition, defamation and transmitting offensive content following the airing of "Locked Up in Malaysia's Lock Down", a documentary on the plight of migrants under Malaysia's so-called Movement Control Order. Al Jazeera's staff were faced with abuse and death threats both offline and online for allegedly sullying Malaysia's image. Similarly, a Bangladeshi national, Mohamad Rayhan Kabir, was arrested and later deported for criticizing the government's treatment of undocumented migrants in an interview. During the pandemic, the government has even issued a gag order against civil servants from sharing online comments that are critical towards the government (Palansamy 2021).

Self-censorship is common in Malaysia, and it can be argued to have been encouraged by the authority as the MCMC released a statement in January 2021 reminding internet users not to post anything that is offensive involving the "3R": Royalty, Religion, and Race (MCMC 2021a). Religion has always been a sensitive topic in Malaysia especially with the rise of Islamist populists who believe in the superiority of Islam and Muslims in Malaysia since the Constitution declares Islam as the religion of the federation. Criticism of Islam is withheld from mainstream media, but dangerous and hateful speech by Islamist populists appear to go unnoticed by the government. Understandably, the government has to walk the tightrope of maintaining peace without upsetting the majority Muslim population.

Individuals are commonly charged and prosecuted under Section 233 of the CMA 1998 or the Sedition Act 1948. Fahmi Reza, a well-known graphic designer, has been investigated at least nine times by the police for his satirical artworks criticizing the government including the royalty. Police officers even entered Fahmi's house by force in April 2021 to arrest him for alleged sedition. The investigations were carried out under Section 4(1) of the Sedition Act and Section 233 of the Communications and Multimedia Act (Lim 2021). As reported by ECPS (Yilmaz et al. 2022), targeted arrests and violence include "influential digital activists and actors being physically assaulted, arrested, and sentenced to jail." (p. 12)

Another recent case involved Ain Husniza, a 17-year-old who exposed a male teacher on TikTok for allegedly making a rape joke in class. Instead of protecting her, Ain was served with a defamation suit and called by the police to make a statement for "breaching the peace" (Goh 2021). At the same time, Ain has been harassed and abused online by those who defended the teacher on grounds that it was simply a 'joke'. Beyond misogyny, there is also an element of religious superiority with online commenters criticising her for not wearing the Islamic headcover or hijab (Al Jazeera 2021).

The Malaysian government under the former prime minister, Muhyiddin Yassin, was criticized in late 2020 for rebranding the Special Affairs Department (JASA) into the Department of Community Communication (J-KOM) with an initial budget of RM85.5 million. Following public outrage, the department's budget was reduced to RM40 million. J-KOM is accused by the opposition of being the government's propaganda machine that is also involved in funding cybertroopers who are paid to create positive contents for the government and to ruthlessly criticize the opposition. The Oxford Internet Institute reported

that cybertroopers in Malaysia use bots to flood social media, spread disinformation and engender further social polarization (Bradshaw et al. 2021). Employing trained cybertroopers, colloquially known as "cytro", is a known strategy in Malaysia among political parties from both sides of the aisle.

As discussed in the literature review, the government has also been attempting to write its own online narrative. However, nefarious ways have been used to achieve this goal. In 2022, the Royal Malaysian Police was accused by Meta, Facebook's parent company, of being linked to a "troll farm" with coordinated efforts to promote the then-government and criticise the opposition (Zolkepli 2022). This is worrying as the police is supposed to be a non-partisan institution that is responsible for public peace. In this way, digital freedom has been taken advantage of by government institutions for authoritarian measures against political opponents. The Royal Malaysian Police strongly denied the accusations (Babulal 2022).

## 4. Religion's Role in Digital Authoritarianism

Religion's role in digital authoritarianism can be ascertained, among other things, by looking at four aspects: religious justifications of internet curbs by the government, prominent role of religious leaders in restricting digital freedom, laws to limit blasphemy, heresy, heterodoxy, and pornography, and more restrictions on religious minorities or in areas where religious minorities live in large numbers.

Malaysia is a multicultural and multireligious country with Malay-Muslims making up a majority. Since Islam has been effectively bureaucratized by the government from the first premiership of Mahathir Mohamad from 1981 until 2003, there has been a rise of Islamisation in the country. As such, conservative Islamic values, especially concerning sexual elements, have always been censored in Malaysia according to the Film Censorship Act 2002. Public screening of films with nudity or sex scenes would render the subject liable to fines and/or imprisonment. Sensitive religious material would also be thoroughly vetted such as Mel Gibson's "The Passion of the Christ" which was barred from being shown to Muslims.

The Malaysian Communication and Multimedia Content Code also specified "indecent content" to include nudity and sex, whereas "obscene content" involve explicit sex acts or pornography, child pornography and sexual degradation. However, with streaming services like Netflix offering subscription to Malaysian viewers, concerns have been raised over the absence of censorship on nudity, sex scenes, and LGBTQ+ content on the platform. This issue reached its peak in 2019 when the National Film Development Corporation (Finas) suggested for the government to regulate Netflix even though censorship is under the jurisdiction of the Film Censorship Board.

Another matter on religious justification for digital authoritarianism is the use of the term 'fitnah' by politicians and the government to condemn fake news and to justify internet curbs especially through self-censorship. Fitnah is basically slander, but in the context of the majority Muslim population in Malaysia, fitnah carries a wider religious connotation. Although fitnah is already part of the Malay language having Arabic origin, the use of the word in Malaysia continues to have Islamic connotation of being sinful. A fitnah is beyond illegal but is also wrong in the eyes of God. Thus, the deliberate use of the word fitnah is to remind Malaysian Muslims of the terrible consequence to befall them if they were to commit slander. In this way, self-censorship becomes widely acceptable by the public. For example, the Qur'anic verse below is used to warn Muslims of this danger of 'fitnah': "Indeed, those who have tortured [fitnah] the believing men and believing women and then have not repented will have the punishment of Hell, and they will have the punishment of the Burning Fire." (Qur'an 85:10)

In 2017 during an intense period before the 2018 general election, the IT Bureau of UMNO, the main Malay party in the country, launched a campaign to fight 'fitnah' on social media. The bureau's chairperson, Ahmad Maslan, reminded the people to report cases of slander to MCMC in order to maintain a conducive, harmonious and peaceful atmosphere

in Malaysia (*Bernama* 2017). Similarly, on the party's mouthpiece, a deputy youth chief of the Islamist PAS party mentioned in relation to social media that "the spread of fitnah [slander] is rampant every time before elections where it can threaten the harmony of people's lives." (Noh 2021) Because of this, when the Islamist PAS was in government, they started the "Stop Fitnah on Social Media" campaign just in time before the 2022 general election. The Malaysian Islamic Development Department uses justification from the Quran and Hadith for their effort to stop the spread of fake news in religious terms (*Malaysiakini* 2022). While regulating misinformation and disinformation is not entirely harmful, it is the political usage of religious terms that is worrying as the boundary becomes blurred.

Religious leaders in Malaysia have usually not taken any firm position regarding restriction of online freedom. They have taken a more neutral position by reminding the people of the danger of spreading lies online even if the words are not uttered or that the victim is a stranger. In other words, self-censorship is very much encouraged to Muslims so as not to invite the wrath of God. For example, the Office of the Mufti of Federal Territory (mufti being a Muslim legal expert) explained that while the usage of social media is generally permissible, it becomes prohibited when the platforms are used for actions such as calling someone a bad name, insulting and degrading others, betraying and lying, and slander and malicious gossip (PMWP 2018).

Perhaps a more direct role of religious leaders can be seen in 2014 when two muftis urged the National Fatwa Council, the country's highest Islamic authority, to release a fatwa (legal opinion on Islam) that prohibits conversation through social media or messaging apps between an unmarried man and woman. According to them, this action may lead to sinful acts and the Malaysian government should emulate a similar fatwa released by Iran's Ayatollah Ali Khamenei (*Malaysiakini* 2014). However, the National Fatwa Council has never released such a fatwa that could restrict male-female conversation on social media.

With Islam being the religion of the federation, Malaysia has strict laws on blasphemy. Articles 295–298A of the Malaysian Penal Code, which is a legacy of British colonial rule, provide penalties for those who insult any religion in Malaysia, although it is usually used to prosecute offenders against Islam. The only recognised Islamic sect in Malaysia is Sunni Islam with Shiites, Ahmadis and those from al-Arqam facing intermittent persecution following their status as "deviant" since 1996. Anyone who is found to have deviated from mainstream Islam would be brought to rehabilitation centres that enforce the government's official version of Islam. Facebook groups actively spread anti-Shia hatred, such as *Gerakan Banteras Syiah* with more than 25,000 followers (Roknifard 2019). It has to be noted, however, that the government had in the past suspended a Tamil-language daily for mistakenly printing an image of Jesus Christ holding a cigarette (Zappei 2007). The blasphemy law thus appears to theoretically apply to offensive acts towards any religion in the country. In cyberspace, these minority groups are relatively free from government interference or persecution.

Moreover, the blasphemy law has been used against the LGBTQ+ community including a prominent transgender entrepreneur in Malaysia, Nur Sajat, who has also been subjected to online harassment for living as an openly transgender woman (Chew 2021). She is charged with blasphemy by the Selangor sharia court for dressing as a woman at an event in 2018. She escaped from the country in early 2021 but was detained by the Thai authorities in September 2021 for an invalid passport and has since been granted asylum in Australia. Although she has been charged for insulting Islam, Nur Sajat is also wanted in Malaysia for failing to pay RM200,000 compensation following a court judgment against her for breach of contract relating to her business (*Malaysiakini* 2021a). Despite harassment by local authorities and social media users, there have been no reported cases of prosecution against Nur Sajat or any LGBTQ+ persons for their online activities. When the minister of religious affairs said in 2018 that the government would set up a regulator to monitor LGBT activity online, nothing came of it (Ellis-Peterssen 2018).

Non-Muslims are prohibited from proselytizing to Muslims in Malaysia. Although it is not illegal according to federal law, most states in Malaysia have specific laws making the act

illegal as jurisdiction over Islamic matters fall under the states. In 2021 the deputy minister in the Prime Minister's Department, Ahmad Marzuk Shaary, said that the government plans to draft new sharia laws including the Control and Restriction on the Propagation of Non-Muslim Religious Bill which is a law that would be applied to the federal territories in Malaysia similar to those already in place in most states (*Malaysiakini* 2021b). Proselytization is such a contentious issue that two persons who have been missing for years were allegedly abducted by the Special Branch of the police force for proselytizing to Muslims. Pastor Raymond Koh who has been missing since 2017 was alleged to have spread Christianity to Muslims while Amri Che Mat who has been missing since 2016 had supposedly spread Shiism to Sunni Muslims in Malaysia.

Malaysia has had some high-profile cases of those who converted from Islam, the most well-known being Lina Joy who exhausted the court process without success to remove Islam from her identification card. Another known convert is Juli Jalaludin. After obtaining a job offer in Norway in June 2013, she departed from Malaysia after being harassed online for joining an apostate group on Facebook. She later started the Facebook pages *Murtad di Pantai Timur* (Apostate in the East Coast) and *Murtad in Kelantan* (Apostate in Kelantan) with a group of her Facebook friends. These pages were later blocked by the government because they were thought to offend and insult Islam (Cheng 2022).

Although Malaysia is a majority Muslim nation, Christians actually make up a majority in the state of Sarawak on Borneo Island, at 61.2%. Its neighbour, the state of Sabah, also has a significant Christian population, at 26.6%. What is more unique about these two states is that the population is made up of multiple indigenous ethnicities that do not belong to a specific religion, in contrast to the Malays in peninsular Malaysia who are constitutionally required to be Muslims. These ethnic groups include the Bajau, Kadazan-Dusun, Iban, Bidayuh and Melanau among others. Among these people, Malay-language has been used in certain Christian congregations and they have used the word "Allah" in Bibles and other religious publications to refer to God. However, since at least 2008, there has been different court cases in which government prosecutors argued the word "Allah" and others such as "al Quran" and "fatwa" are restricted for Muslim usage in Malaysia (US Department of State 2021) in accordance with a circular by the Home Ministry some three decades ago in 1986. It has been argued that this restriction impedes on citizen's rights to freedom of speech and religion. In March 2021, the High Court decided that a Malaysian Christian has the constitutional right to use the word "Allah", "Kaabah", "Baitullah" and "solat" as long as there is a disclaimer the publication is meant for Christians (Anand 2021). Following the verdict, right-wing groups have vowed to fight and protest the court's decision. It was also following this decision that the government decided to introduce new draft bills and amendments to sharia laws that were mentioned in a previous section.

## 5. Online Religious Populism in 2022

After decades of mild digital authoritarianism by the Malaysian government in the name of preserving peace among the Muslims, Malaysia's level of democracy has fluctuated within the realm of flawed democracy. In such a space of weakened institutions, Islamist populists in the form of leaders from PAS took advantage of the situation to spread their ideas. If these Islamist populists were in the opposition in 2018–2020, they were part of the government from 2020 until 2022. Due to political wrangling, parliament was dissolved in October 2022 followed by a general election in November. PAS, which is part of the Perikatan Nasional coalition was initially dismissed from being a serious contender in the 2022 general election that was thought to pit Pakatan Harapan against Barisan Nasional. However, with their massive presence online especially on TikTok, PAS and Perikatan Nasional were able to win over support in Malay heartlands. Although the democratisation of the internet is lauded for allowing for greater participation, it also has the potential to be abused by extremist and violent populist groups that seek to keep certain "others", i.e., the non-Muslims, away from the government.

As part of the government in 2020 until 2022, PAS did not push through its shariasation agenda. But with the looming election, there was a rise of Islamist populist rhetoric as the party sought to carve out its influence. For example, in August 2022, Hadi Awang, the president of PAS and former special ambassador to the Middle East, said that the majority of those responsible for destroying Malaysia's politics and economy were non-Muslims, implying that the minority Chinese were to be blamed for corruption in the country (Free Malaysia Today 2022a). After receiving multiple complaints, the police investigated Hadi Awang under the Penal Code on remarks encouraging harm and incitement as well as the Communications and Multimedia Act's prohibition on abuse of network services. Despite its slanderous populist rhetoric that fits the definition of fitnah discussed earlier, the investigation did not go any further than taking statements. In contrast, when the Deputy President of the opposition Parti Keadilan Rakyat exposed alleged corrupt practices by government officials, the MCMC and police were quick to take his phone away (Free Malaysia Today 2022b). Since Islam is such a powerful force in Malaysia, not only is it utilized to silence opposition, but it is also used to protect Islamist populists in the country.

Following the election on 19 November 2022, Malaysians learnt that PAS had actually made inroads across the country and won 49 seats in parliament (22 per cent). In trying to understand the appeal of the party, there is no denying the success of its concerted effort to paint its opposition as "evil" in the language of the populist. PAS' hateful and dangerous speech has always been targeted against the Democratic Action Party which it previously accused of being a Chinese chauvinist party for its call for equality among Malaysians regardless of race and religion. During the 2022 election period, PAS accused the DAP of being pro-communist and thus anti-Islam as a way to scare Muslims from voting for the party and its Pakatan Harapan coalition (Daim 2022). In a last-minute effort to galvanize the grassroots, the former prime minister, Muhyiddin Yassin, accused Christians and Jews of attempting to Christianise Malaysia (The Star 2022). Civilisational populism is about delegitimising "others" for not belonging to the so-called correct and proper civilisation and in Malaysia's case that means the Islamic *ummah*.

There is no denying that digital authoritarianism in the form of Islamist populism by former members of the government has been rising. It has polarized the nation as seen from the result of the 2022 general election in which most votes were split between right-wing Perikatan Nasional and center-left Pakatan Harapan. Although free speech has to be protected, dangerous speech is an altogether different form of online activity that must be monitored as its effect might spill over from a person's screen and to the streets. When religion is added to the equation to justify this form of dangerous speech it limits a person's liberty in a conservative nation such as Malaysia. Most importantly, it limits the government's ability to stop violent populism on its collision course against democracy. Instead of extending the government's reach through additional laws which might further deteriorate Malaysia's flailing democracy, it is proposed for the government to take measures such as public campaigns to educate the masses on detecting disinformation. Limiting online freedom might lead down a path of digital authoritarianism and hence it is not recommended if Malaysia wishes to move towards democratic consolidation. Instead, the government should empower the masses in making the best decisions, in line with the tenets of liberal democracy.

## 6. Conclusions

This article looks at the use of religion to justify a form of digital authoritarianism in Malaysia that has led to the rise of Islamic populism. This is possible under the banner of protecting the 3R—Religion, Race, and Royalty. From the discussion, it is clear that Malaysia does put certain limitations on digital media in the form of outright blocks to websites, and in more extreme cases, harassment and intimidation of individuals and political opponents. However, this form of digital oppression in Malaysia has been based on the political manoeuvring of the government against alleged sedition more than on religious grounds. Although Islamist populism has been on the rise since 2018, religion

has seldom been used to justify any kind of blanket internet or social media censorship except when it comes to issues such as pornography and gambling. Malaysian cyberspace is relatively free with members of different communities evading persecution for their online activities. Since political and religious figures are both relatively active on social media, rather than decrying the platform as 'evil', religious figures have mostly come up with advice to be cautious when posting information online as spreading lies is sinful in Islam under the concept of fitnah.

Due to this relative openness, and the fact that the government has never imposed a total shutdown of the internet, Malaysia's Freedom on the Net score has been relatively stable (Table 1) although categorised as partly free. Wherever religion is used to justify censorship, it is against "deviant" groups such as Shia Islam and Ahmadiyya or those who were proselytising to Muslims. In a multicultural country where Islam is highly institutionalised, the fragility of social cohesion is of the utmost concern to the government and if maintaining social cohesion entails utilising authoritarian methods such as controlling sub-networks, proxies, and network-nodes, the government sees it only as necessary. However, with the polarising election campaign in 2022, Islamist populists who used to be part of the government have taken advantage of the open space to spread disinformation that is couched in religious rhetoric. This is another form of digital authoritarianism as it threatens democracy within Malaysia by delegitimising the non-Muslim "others". Since Malaysia has experienced changes in government, it is to be seen if the forces of democracy within civil society will be strong enough to counter rising populist authoritarianism on the internet.

**Funding:** This research received no external funding.

**Institutional Review Board Statement:** Not applicable.

**Informed Consent Statement:** Not applicable.

**Data Availability Statement:** No new data were created or analysed in this study.

**Conflicts of Interest:** The author declares no conflict of interest.

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
