# Peer review of "Digital Authoritarianism: Protecting Islam in Multireligious Malaysia"

_religions, doi:10.3390/rel14010087_

Round 1

Reviewer 1 Report

Overall I think this manuscript is fine.  My one complaint would be that it isn't really situated in a theoretical or thematic argument.  It is basically a straight-forward report on empirical details related to Malaysian populist Islam and how this is tied to institutional politics and the regulation of the internet.  It is a fine example of this, but this reader was hoping for more.

One way to provide more might be to focus a bit on the content of the Islamic populism described in the paper.  For example, the author seems to imply that the term "fitnah," which becomes central to the analysis beginning on page 8, has a meaning that is deeply resonate with Islam.  If so, bring that out a bit.  In Indonesian (the form of Malay with which I am most familiar), this isn't necessarily the case.  Fitnah, of course, comes from Arabic/Islamic contexts, but at this point it is fully Indonesian and is used in a way that isn't really informed in any deep way by its Islamic context.  If this is different in Malaysia, it might be worth pulling that out here, to make it clear to the reader how Islamic populism is encoded in everyday political speech.

Doing more of this kind of thing, and thereby bringing out the texture of Islamic populism in Malaysia, would take an okay/fine article, and potentially make it an interesting/great article.

One last point: The shift in focus from Malaysian control of political speech and the internet to the free reign that Islamic populists have on online platforms (and the call to have this regulated) does make sense to me, especially as an American who shares similar concerns over right-wing discourses coming from electronic media in my own country.  That said, the shift back-and-forth between these two positions demands more theorization, especially because in one case the author is suggesting less regulation/interference, but more in the other.  What theoretically links these different cases that demands these seemingly opposite responses?  I am presuming it is anti-democratic (and sometimes violent) populism.  If that is the argument, do not leave it implied; but emphasize it.

Author Response

Overall I think this manuscript is fine.  My one complaint would be that it isn't really situated in a theoretical or thematic argument.  It is basically a straight-forward report on empirical details related to Malaysian populist Islam and how this is tied to institutional politics and the regulation of the internet.  It is a fine example of this, but this reader was hoping for more. That said, the shift back-and-forth between these two positions demands more theorization, especially because in one case the author is suggesting less regulation/interference, but more in the other.  What theoretically links these different cases that demands these seemingly opposite responses?  I am presuming it is anti-democratic (and sometimes violent) populism.  If that is the argument, do not leave it implied; but emphasize it.

Thank you for the valuable comment. I have added a discussion on the democratic recession and how government’s digital authoritarianism has led to democratic backsliding. Thus, it becomes the perfect opportunity for populists to appear and delegitimize political opponents.

One way to provide more might be to focus a bit on the content of the Islamic populism described in the paper.  For example, the author seems to imply that the term "fitnah," which becomes central to the analysis beginning on page 8, has a meaning that is deeply resonate with Islam.  If so, bring that out a bit.  In Indonesian (the form of Malay with which I am most familiar), this isn't necessarily the case.  Fitnah, of course, comes from Arabic/Islamic contexts, but at this point it is fully Indonesian and is used in a way that isn't really informed in any deep way by its Islamic context.  If this is different in Malaysia, it might be worth pulling that out here, to make it clear to the reader how Islamic populism is encoded in everyday political speech. Doing more of this kind of thing, and thereby bringing out the texture of Islamic populism in Malaysia, would take an okay/fine article, and potentially make it an interesting/great article.

I have tried my best to flesh out further the discussion on Islamist populism in Malaysia.

Reviewer 2 Report

P. 1. There are minor English errors such as „ the removal inflammatory content”

p. 1. The author shows subjective political opinions „ Unfortunately, the Pakatan Harapan experiment lasted for a mere 22”. This does not help the non-affiliated reader to understand the context and the involved actors. The author should moderate the expression of his/her political views.

P1. the author takes uncritically Freedom House, Corruption Perception Index, and World Press Freedom Index as criteria of free press.

P2. the quote which ends with “to prevent independent outlets from covering key events.” is not referenced.

p2. the style is repetitive „Next, we will discuss the various methods of government restrictions online and finally a discussion on the role of religion”

p2. It is irrelevant to discuss the South East Asian context. Malaysia is the proper context of the study and the data cannot be generalisable to South East Asia. There is no comparison to be made between these various countries. Please delete most of pages 2 and 3 on South East Asia and focus on Malaysia.

p.5 “Unlike countries such as Turkey, India and Pakistan, the Malaysian government has 224 yet to fully shut down the internet”. When did Turkey, India and Pakistan shut down the internet?

P10. the section on Online religious populism in 2022 does not fit in the article’s scope.

p. 11 the author mixes islamist rhetoric and the government’s policies. Please focus on the latter as indicated in the title of the article.

P. 11 also contains controversial statements such as “the Malay psyche”. Please rewrite this page. The whole discussion on the 2022 elections is bitter.

Author Response

1. There are minor English errors such as „ the removal inflammatory content”

Fixed.

2. The author shows subjective political opinions „ Unfortunately, the Pakatan Harapan experiment lasted for a mere 22”. This does not help the non-affiliated reader to understand the context and the involved actors. The author should moderate the expression of his/her political views.

More context has been provided.

P1. the author takes uncritically Freedom House, Corruption Perception Index, and World Press Freedom Index as criteria of free press.

The data is supported by a report from Malaysia’s own think tank to provide further evidence.

P2. the quote which ends with “to prevent independent outlets from covering key events.” is not referenced.

Added cited paragraph for more clarification.

p2. the style is repetitive „Next, we will discuss the various methods of government restrictions online and finally a discussion on the role of religion”

The repeated word has been deleted.

p2. It is irrelevant to discuss the South East Asian context. Malaysia is the proper context of the study and the data cannot be generalisable to South East Asia. There is no comparison to be made between these various countries. Please delete most of pages 2 and 3 on South East Asia and focus on Malaysia.

The section on Southeast Asia has been deleted and the focus now is on democracy and digital authoritarianism in Malaysia.

p.5 “Unlike countries such as Turkey, India and Pakistan, the Malaysian government has 224 yet to fully shut down the internet”. When did Turkey, India and Pakistan shut down the internet?

Examples of internet shutdown in these countries were provided.

P10. the section on Online religious populism in 2022 does not fit in the article’s scope. The author mixes islamist rhetoric and the government’s policies. Please focus on the latter as indicated in the title of the article.

Since reviewer 1 sees value in this section, I have tried my best to explain the connection between the government’s digital policies that had allowed for the rise of online Islamist populism in 2022 by former government officials.

P11 also contains controversial statements such as “the Malay psyche”. Please rewrite this page. The whole discussion on the 2022 elections is bitter.

The section has been re-written to avoid further bias. Thank you.